# Transient Changes in the Plasma of Astrocytic and Neuronal Injury Biomarkers in COVID-19 Patients without Neurological Syndromes

**DOI:** 10.3390/ijms24032715

**Published:** 2023-02-01

**Authors:** Matthew P. Lennol, Nicholas J. Ashton, Oscar Moreno-Pérez, María-Salud García-Ayllón, Jose-Manuel Ramos-Rincon, Mariano Andrés, José-Manuel León-Ramírez, Vicente Boix, Joan Gil, Kaj Blennow, Esperanza Merino, Henrik Zetterberg, Javier Sáez-Valero

**Affiliations:** 1Instituto de Neurociencias de Alicante, Universidad Miguel Hernández-CSIC, 03550 San Juan de Alicante, Spain; 2Centro de Investigación Biomédica en Red Sobre Enfermedades Neurodegenerativas (CIBERNED), 03550 San Juan de Alicante, Spain; 3Department of Psychiatry and Neurochemistry, Institute of Neuroscience and Physiology, The Sahlgrenska Academy at the University of Gothenburg, 431 80 Mölndal, Sweden; 4Centre for Age-Related Medicine, Stavanger University Hospital, 4005 Stavanger, Norway; 5Department of Old Age Psychiatry, Institute of Psychiatry, Psychology and Neuroscience, King’s College London, London SE5 8AF, UK; 6Biomedical Research Unit for Dementia at South London and Maudsley NHS Foundation, NIHR Biomedical Research Centre for Mental Health, London WC2R 2LS, UK; 7Departmento de Endocrinología y Nutrición, Hospital General Universitario Dr. Balmis, 03010 Alicante, Spain; 8Instituto de Investigación Sanitaria y Biomédica de Alicante (ISABIAL), General University Hospital of Alicantet, 03010 Alicante, Spain; 9Departmento de Medicina Clínica, Universidad Miguel Hernández de Elche, 03550 Alicante, Spain; 10Unidad de Investigación, Hospital General Universitario de Elche, FISABIO, 03203 Elche, Spain; 11Departmento de Medicina Interna, Hospital General Universitario Dr. Balmis, 03010 Alicante, Spain; 12Departamento Reumatología, Hospital General Universitario Dr. Balmis, 03010 Alicante, Spain; 13Departmento Neumología, Hospital General Universitario Dr. Balmis, 03010 Alicante, Spain; 14Unidad de Enfermedades Infecciosas, Hospital General Universitario Dr. Balmis, 03010 Alicante, Spain; 15Clinical Neurochemistry Laboratory, Sahlgrenska University Hospital, 413 45 Mölndal, Sweden; 16Department of Neurodegenerative Disease, UCL Institute of Neurology, London WC1E 6BT, UK; 17UK Dementia Research Institute, King’s College London, London WC1E 6BT, UK; 18Hong Kong Center for Neurodegenerative Diseases, Hong Kong, China; 19UW Department of Medicine, School of Medicine and Public Health, Madison, WI 53726, USA

**Keywords:** biomarker, COVID-19, GFAP, NfL, plasma, T-tau

## Abstract

The levels of several glial and neuronal plasma biomarkers have been found to increase during the acute phase in COVID-19 patients with neurological symptoms. However, replications in patients with minor or non-neurological symptoms are needed to understand their potential as indicators of CNS injury or vulnerability. Plasma levels of glial fibrillary acidic protein (GFAP), neurofilament light chain protein (NfL), and total Tau (T-tau) were determined by Single molecule array (Simoa) immunoassays in 45 samples from COVID-19 patients in the acute phase of infection [moderate (n = 35), or severe (n = 10)] with minor or non-neurological symptoms; in 26 samples from fully recovered patients after ~2 months of clinical follow-up [moderate (n = 23), or severe (n = 3)]; and in 14 non-infected controls. Plasma levels of the SARS-CoV-2 receptor, angiotensin-converting enzyme 2 (ACE2), were also determined by Western blot. Patients with COVID-19 without substantial neurological symptoms had significantly higher plasma concentrations of GFAP, a marker of astrocytic activation/injury, and of NfL and T-tau, markers of axonal damage and neuronal degeneration, compared with controls. All these biomarkers were correlated in COVID-19 patients at the acute phase. Plasma GFAP, NfL and T-tau levels were all normalized after recovery. Recovery was also observed in the return to normal values of the quotient between the ACE2 fragment and circulating full-length species, following the change noticed in the acute phase of infection. None of these biomarkers displayed differences in plasma samples at the acute phase or recovery when the COVID-19 subjects were sub-grouped according to occurrence of minor symptoms at re-evaluation 3 months after the acute episode (so called post-COVID or “long COVID”), such as asthenia, myalgia/arthralgia, anosmia/ageusia, vision impairment, headache or memory loss. Our study demonstrated altered plasma GFAP, NfL and T-tau levels in COVID-19 patients without substantial neurological manifestation at the acute phase of the disease, providing a suitable indication of CNS vulnerability; but these biomarkers fail to predict the occurrence of delayed minor neurological symptoms.

## 1. Introduction

The coronavirus disease 2019 (COVID-19), which is caused by severe acute respiratory syndrome coronavirus 2 (SARS-CoV-2) infection, includes the occurrence of neurological symptoms, mainly in patients with severe COVID-19 [1]. Whether COVID-19 neurological manifestations are a primary consequence of SARS-CoV-2 brain infections or is mediated by the neuroinflammatory responses remains unclear.

In this regard, several manuscripts have demonstrated altered levels of several glial and neuronal cerebrospinal (CSF) and plasma biomarkers in COVID-19 patients with neurological symptoms [2,3,4,5,6,7,8,9]. Most of these studies have addressed the levels of the neurofilament light chain (NfL) and glial fibrillary acidic protein (GFAP), which are markers of neuroaxonal damage and astrocytic activation/injury, respectively. However, whilst some reports have also shown altered levels of plasma NfL among mildly symptomatic or asymptomatic adult patients [10], others have failed to find changes in children with asymptomatic to moderate COVID-19 [11].

Herein, we determined whether the plasma levels of NfL and total tau (T-tau, another marker of axonal damage and neuronal degeneration), as well of GFAP, were altered in COVID-19 patients without major neurological manifestations, as compared with non-infected and neurologically healthy controls. We also analyzed whether plasma levels of these biomarkers are restored in patients after a recovery period. Furthermore, we assessed whether the levels of circulating angiotensin-converting enzyme 2 (ACE2) species, the host receptor of SARS-CoV-2, showed any changes in COVID-19 patients.

## 2. Results

Table 1 shows the clinical and demographic characteristics of the COVID-19 group. No significant differences in age were found between infected (n = 45) and non-infected subjects (n = 14). Concentrations of neuronal and glial biomarkers in the different groups are given in Figure 1. Patients with COVID-19 had significantly higher plasma concentrations of GFAP (160%; *p* < 0.001), NfL (100%; *p* = 0.015) and T-tau (195%; *p* = 0.001) than controls. No significant differences were found for any biomarker between patients with moderate or severe presentation of COVID-19 (*p* values between, *p* = 0.320–0.989; see Figure 1). In COVID-19 patients, a correlation was found between the levels of neuroaxonal damage biomarkers, NfL and T-tau (r = 0.583; *p* < 0.001), but also between NfL and GFAP (r = 0.671; *p* < 0.001) and between T-tau and GFAP (r = 0.667; *p* < 0.001) (Figure 2). None of these correlations were significant in controls. GFAP (r = 0.585; *p* < 0.001), NfL (r = 0.637; *p* < 0.001) and T-tau (r = 0.403; *p* = 0.007) were significantly correlated with age for patients with COVID-19, but not in controls. No significant differences were observed when the COVID-19 or control subjects were sub-grouped by gender. We were not able to identify an association between any biomarker determined at the acute phase of the disease with medium-term persistence of asthenia, myalgia/arthralgia, anosmia/ageusia, vision impairment, headache, memory loss or other post-COVID symptoms. GFAP concentration correlated with troponin (r = 0.652; *p* < 0.001) and the B-type natriuretic peptide (BNP; r = 0.500; *p* = 0.002), and the same occurred with NfL (troponin: r = 0.728; *p* < 0.001; BNP: r = 0.629; *p* < 0.001) and T-tau (troponin: r = 0.609; *p* < 0.001; BNP: r = 0.386; *p* = 0.017). NfL also displayed a correlation with levels of the D-dimer (r = 0.328; *p* = 0.037) and ferritin (r = 0.354; *p* = 0.019), and T-tau correlated negatively with C-reactive protein (CRP; r = −0.313; *p* = 0.039). There was no significant correlation with lactate dehydrogenase (LDH), IL-6 concentrations or with the number of lymphocytes.

The clinical follow-up provided a suitable reference to assess full recovery after a period of 58–70 days. Levels of GFAP, NfL and T-tau levels were restored after ~2 months of recovery in moderate and severe COVID-19 patients (Figure 1), and there were no significant differences between patients with and without neurological symptoms as part of post-acute infection sequelae. In paired samples obtained from the same patients (n = 11), these results were confirmed, with a significant decrease in the levels of GFAP (*p* = 0.032), NfL (*p* = 0.005), and a trend to decrease for T-tau levels (*p* = 0.053), when comparing levels at the acute phase with the recovered stage, as determined by the Wilcoxon Signed Rank test.

ACE2 is the host receptor of the SARS-CoV-2 virus, and ACE2 fragments are shed after interaction with the SARS-CoV-2 spike (S) protein resulting in ~20–25 kDa smaller fragments than the original full-length species [12]. We have recently demonstrated the co-existence of several soluble ACE2 full-length species, as well as cleaved fragments, in human plasma [13]. The cleaved ACE2 fragment would increase at the expense of the membrane resident full-length ACE2, with soluble circulating full-length species probably reflecting the tissue content of ACE2. Since detection of viral RNA or positive staining did not seem to correlate with the extent of (neuro)pathological changes, nor active infectivity of particular tissues, we used the determination of circulating ACE2 species as biochemical parameters to assess active systemic SARS-CoV-2 infection. Due to the co-existence of different circulating ACE2 species, serving as a read out of different dynamics during disease progression, here we analyzed plasma samples by Western blotting using either a polyclonal goat antibody (AF933) that recognizes the ectodomain of ACE2, or a rabbit polyclonal antibody (ab15348) raised against the C-terminus of human ACE2, thus detecting only full-length species and not C-terminally truncated fragments. The combined use of both antibodies confirmed our previous report [13] and the existence of 70 and 75 kDa fragments, and the full-length species ranking from 95 to 170 kDa (Figure 3A). Remarkably, in the COVID-19 samples, the 70 kDa ACE2 cleaved fragment displayed a tendency to increase (42%; *p* = 0.09), compared with controls, which may represent virus entry and subsequent proteolytic processing of the membrane receptor (Figure 3B). The 75 kDa fragment levels were unchanged (Figure 3C), which is also consistent with previous reports, thus indicating that the 70 kDa fragment is the only one associated with SARS-CoV-2 infection. The 70 kDa fragment is compatible in molecular mass with the expected shed fragment originating from 95–100 kDa full-length species. The circulating 95–100 kDa full-length species decreased significantly (62%; *p* = 0.002) in COVID-19 patients compared with controls (Figure 3D). An ACE2 “cleaved/full-length” quotient [70 kDa/(95 + 100 kDa)] appears to be a better marker for discriminating between COVID-19 and controls (Figure 3E). The decrease in 95–100 kDa full-length species of ACE2 was also correlated significantly with GFAP (r = 0.31; *p* = 0.043), T-tau (r= 0.46; *p*= 0.002) and was close to achieving significance for NfL (r = 0.30; *p* = 0.051). Similar trends to decrease were observed for other soluble ACE2 full-length species present in plasma (Figure 3F–G), which also displayed correlations with CNS injury biomarkers.

Levels of circulating ACE2 full-length species were also in the normal range in patients after this recovery period (Figure 3D,F,G), serving also as a biochemical biomarker to substantiate recovery.

## 3. Discussion

Earlier studies indicate that minor neurological manifestations, such as headaches and dizziness, are present in COVID-19 patients during the acute infection or after a delay [14]; but with the exponential progress of the pandemic, reports of neurological manifestations are also increasing, indicating the potential occurrence of severe neurological injury [1,15,16]. Indeed, SARS-CoV-2 has been shown to have the capability to enter the brain [17]; but only a small percentage of the PCR-diagnosed cases showed threshold amplification in the brain, even though some displayed neuropathological findings that could be unequivocally attributed to COVID-19 disease [18]. PCR for SARS-CoV-2 also gave a negative result in the cerebrospinal fluid of COVID-19-related encephalitis, strongly suggesting a cytokine-induced neuroinflammation as the underlying mechanism of SARS-CoV-2 related encephalitis [19]. Thus, it is unclear whether the mild and severe neurological features of COVID-19 are mainly a consequence of distress and multiorgan failure causing neuroinflammation and are not directly due to brain infection (discussed in [20]).

Neurological involvement in systemic diseases is frequently associated with adverse effects on morbidity and mortality, and preventing the impact of SARS-CoV-2 infection on the nervous system is of high interest. Thus, in parallel to the increasing evidence of neurological symptoms in COVID-19 and the pathophysiology behind these manifestations, a number of studies have addressed whether circulating CNS injury biomarkers could be prognostic markers. Most of these studies on plasma glial and neuronal biomarkers were performed in neurologically affected COVID-19 patients in the acute phase of the disease. Thus, early studies demonstrated that patients with severe COVID-19 had higher plasma concentrations of GFAP and NfL than controls, while GFAP was also increased in patients with a moderate manifestation of the disease [2]. Elevated serum/plasma NfL and GFAP levels have been consistently found across patients hospitalized with COVID-19, in a severity-dependent manner [21], particularly associated with an unfavorable short-term outcome [3,22,23,24]. In severely affected cases, NfL, GFAP, and also T-tau were significantly increased in patients with a fatal outcome [5,6]. Again, some studies suggest that in severe COVID-19, the central neuronal and axonal damage in these patients may be driven, in part, by the level of systemic cardiovascular disease and peripheral inflammation [25]. In any case, altogether, these studies indicate that blood concentrations of GFAP, NfL and T-tau increase with disease severity during the acute phase of COVID-19.

However, in this study, we demonstrate significantly elevated plasma concentrations of GFAP, NfL and T-tau in patients with COVID-19, but with non-substantial neurological affectation, compared with controls. Previous reports also found higher baseline serum NfL levels in COVID-19 patients without major neurological symptoms, compared with controls, associated with worse clinical outcomes [26] or regardless of disease severity [10]. The concentrations of NfL also increased over repeated measurements in non-survivors, whereas in survivors the levels remained stable [4]. Moreover, elevated GFAP has been associated with disease severity, but not irrespective of neurological manifestations [8,27].

In our study, all the CNS damage biomarkers correlated amongst themselves, but failed to indicate differences when post-COVID patients with minor neurological manifestations were sub-grouped. In any case, the dynamics of each biomarker could be different. An early study demonstrated that after a short follow-up of 11.4 days, patients with severe COVID-19 displayed a decrease in the earliest plasma GFAP peak, whilst NfL showed a sustained increase [2], perhaps reflecting a sequence of early astrocytic response and more delayed axonal injury. A distinct process characterized by elevation of serum T-tau was seen in patients at follow-up, which appeared to be independent of initial disease severity and was not associated with dysregulated immune responses, unlike NfL and GFAP [21]. Nonetheless, these particular dynamics should be considered in the context of changes that reflect neurological features, and not basal changes occurring during COVID-19 unrelated with neurological manifestations.

The increases in CNS damage biomarkers only show a weak correlation with some hemaetological parameters, including the inflammatory-associated troponin and BNP, and partially D-dimer and ferritin; but T-tau correlated negatively with CRP and no correlations were found with the pro-inflammatory cytokine IL-6. COVID-19-associated neuroinflammation has been extensively associated with a cytokine storm, particularly in severe COVID-19 [28,29]. In this context, previous studies of patients with COVID-19 at acute phase demonstrated from elevations in serum of NfL and GFAP also associated with elevations of pro-inflammatory cytokines [21]. Elevated IL-6, IL-18 and IL-8 levels have been also described in the serum and CSF of COVID-19 patients in parallel with elevated serum NfL, but not with CSF NfL [30]. Other studies demonstrated that NfL concentrations were not correlated with CRP or ferritin, often found to be associated with hyperinflammation in COVID-19 patients, suggesting that the increased NfL concentrations merely reflect enhanced inflammation [4]. Again, given the particular dynamics of systemic inflammatory biomarkers, it can be difficult to establish associations with the changes in CNS damage biomarkers.

This study also demonstrated the normalization of plasma GFAP, NfL and T-tau levels 2 months after recovery in patients with or without minor post-acute neurological sequelae. A previous longitudinal study also demonstrated that after six months, NfL and GFAP concentrations—which were elevated in the acute phase of patients suffering COVID-19—returned to normal [31]. In this previous study, despite the normalization of plasma concentrations of CNS injury biomarkers, a significant number of patients still experienced persistent neurological and cognitive symptoms, and no association was found between higher concentrations of CNS injury biomarkers during the acute phase and post-infectious neurological symptoms [32], which is consistent with our results.

Interestingly, in our study, plasma GFAP, NfL and T-tau were also correlated with the altered quotient between ACE2 cleaved fragments and circulating full-length species that probably reflect enhanced SARS-CoV-2 interaction with the membrane-resident receptor and further proteolytic cleavage [32], which were seen to be affected in the previous analysis of the same cohort [13]. ACE2 is also present in the brain [33,34,35] and human neurons are indeed a target for SARS-CoV-2 [36]. However, these decreases in circulating ACE2 full-length species noticed in moderate COVID-19 patients reflect the generalized SARS-CoV-2 infection and no particular changes in the CNS, since these decreases affect all the circulating full-length ACE2 species, each of which probably reflects a particular contribution of different tissues/cells expressing different ACE2 isoform patterns [13]. Restoration of the “ACE2 cleaved/full-length” quotient after 2 months of recovery could also serve to assess recovery.

This study has several limitations, with the most obvious derived from the limited size of the cohort and the lack of inclusion of samples from patients with major neurological symptoms. The samples were obtained during the first wave of the COVID-19 pandemic in 2020, where neurological features may have been underestimated; nonetheless, this is a well-characterized cohort treated and followed by a multidisciplinary clinical team.

In conclusion, our data, together with previous reports, served to demonstrate that, when compared with control levels, increased plasma GFAP, NfL and T-tau in COVID-19 patients in the acute phase of infection are indicative of CNS vulnerability. However, altered levels of plasma CNS-injury biomarkers in the acute phase of the disease occurs in COVID-19 patients without delayed neurological manifestation, and, therefore, cannot be considered in itself as informative of existing neurological features. Accordingly, potential cut-offs for considering these biomarkers as indicative of major neurological manifestations in severely affected cases should be estimated by comparing values in COVID-19 patients with and without major neurological symptoms, and not with respect to values determined in non-disease control cases.

## 4. Methods

### 4.1. Plasma Samples

Samples and data from patients included in this study were provided by the BioBank ISABIAL, integrated with the Spanish National Biobanks Network and with the Valencian Biobanking Network, and were processed following standard operating procedures with the appropriate approval of the Ethical and Scientific Committees of the Hospital General Universitario de Alicante (HGUA)-ISABIAL and UMH.

Inclusion criteria: all samples were obtained during the year 2020, between the months of March and June (first wave in Spain) in patients with the following: a positive reverse transcription polymerase chain reaction (RT-PCR) test for SARS-CoV-2 on nasopharyngeal swabs; availability of viable bioBank samples; and clinical data records in the electronic medical system. Samples of forty-five COVID-19 hospitalized patients (n = 10 severe and n = 35 moderate) in the acute phase of infection (less than 14 days of clinical evolution) were included (18 females/27 males; mean age± SEM (range): 64 ± 3 (21–89)years). Hospitalized survivor patients (severe pneumonia) and mild pneumonia patients that were followed-up at home were offered an assessment by a COVID-19 medical team 10–14 weeks after ambulatory COVID-19 recovery or discharge from hospital. Twenty-six plasma samples were collected after recovery from COVID-19 patients with an interval of (mean ± SEM) 63 ± 1 ((range) 58–70) days from the onset of symptoms. Eleven of the recovery samples were from patients with samples in the acute phase.

Thirty-five patients infected with SARS-CoV-2 suffered a moderate presentation of COVID-19 (WHO ordinary scale 3–5), but 10 cases were considered as severe since they suffered from respiratory failure requiring invasive mechanical ventilation and/or were treated at an intensive care unit (ICU) (WHO ordinary scale ≥ 6). Some of the patients suffered delayed neurological symptoms (re-evaluation after 3 months of the acute episode) such as headache or brief and transitory memory loss; none suffered severe neurological features. For clinical and demographic details, and hematological parameters of the COVID-19 patients, see Table 1. Part of this cohort was previously examined in a previous report [13].

A non-disease, non-infected control group, obtained prior to the COVID-19 pandemic, was also analyzed (n = 14; 7 females/7 males; 61 ± 1 [52–68] years).

Blood samples were collected in EDTA tubes, and plasma was obtained by centrifugation at 3000× *g* for 15 min at 4 °C, and then aliquoted and frozen at −80 °C until use.

### 4.2. Biomarker Measurements

All plasma GFAP, tau and NfL measurements were performed at the Clinical Neurochemistry Laboratory at Sahlgrenska University Hospital by board certified laboratory technicians blinded to clinical data using the Neurology 4-PLEX A assay run on an HD-X Analyzer, as described by the manufacturer (Quanterix, Billerica, MA, USA). A single batch of reagents was used; inter- and intra-assay coefficients of variation were below 15% for all analytes.

ACE2 species were detected by fluorescent-based imaging after sodium dodecyl sulfate-polyacrylamide gel electrophoresis (SDS-PAGE) and Western blotting. Plasma samples were heated in reducing Laemmli SDS sample buffer (Thermo ScientificTM, Massachusetts, USA) for 7 min at 70 °C (dilution ratio 1:10). Plasma samples (0.4 μL loaded) were then resolved on 7.5% SDS-PAGE gels (Mini-PROTEAN^®^ TGX™ Precast Gels; Bio-Rad, Munich, Germany) and transferred to 0.2 μm nitrocellulose membranes (Bio-Rad). Then, the membrane was blocked with Odyssey Blocking Buffer (PBS) and incubated with the anti-ectodomain AF933 antibody (1:200 dilution) or alternatively with the anti-C-terminus ab15348 antibody (1:500 dilution). Finally, blots were washed and incubated with the conjugated secondary antibodies (IRDye 800CW donkey anti-goat and IRDye 680 RD goat anti-rabbit, LI-COR Biosciences, Lincoln, USA) and imaged on an Odyssey Clx Infrared Imaging System (LI-COR Biosciences). A control plasma sample was used to normalize the immunoreactive signal between blots. All samples were analyzed at least in duplicate. Band intensities were analyzed using LI-COR software (Image Studio Lite). During preliminary analysis, to rule out the possibility that differences in ACE2 levels were due to loading inaccuracies, following electrophoresis, the gel was divided into two parts, one for protein visualization by SimplyBlueTM SafeStain Coomassie (ThermoFisher Scientific, cat# LC6060) and one for blotting with the AF933 antibody.

To estimate the relative “cleaved/full-length” quotient of ACE2 species (for each sample), the immunoreactivity was considered for the 70 kDa fragment and the sum of the 95 and 100 kDa species.

### 4.3. Statistical Analysis

All data were analyzed using SigmaStat (Version 3.5; SPSS Inc., Chicago, IL, USA). The Kolmogorov–Smirnov test was used to analyze the distribution of each variable. ANOVA was used for parametric variables and the Kruskal–Wallis test for non-parametric variables for comparison between groups. A Student’s *t*-test for parametric variables and a Mann–Whitney U test for non-parametric variables (most of them) were employed for comparison between two groups, and for determining *p* values. For the pair results from repeated measurements on the same COVID-19 patients, at the acute phase and at recovery, the Wilcoxon Signed Rank Test was employed. U-Mann–Whitney was also used to assess whether the levels of markers of neuronal damage during the acute episode of infection were different in the patients that suffered neurological sequelae in the medium term. For correlations, the Rho Spearman test was used. Correlation analysis was performed including total concentration levels of plasma biomarkers.

## Figures and Tables

**Figure 1 ijms-24-02715-f001:**
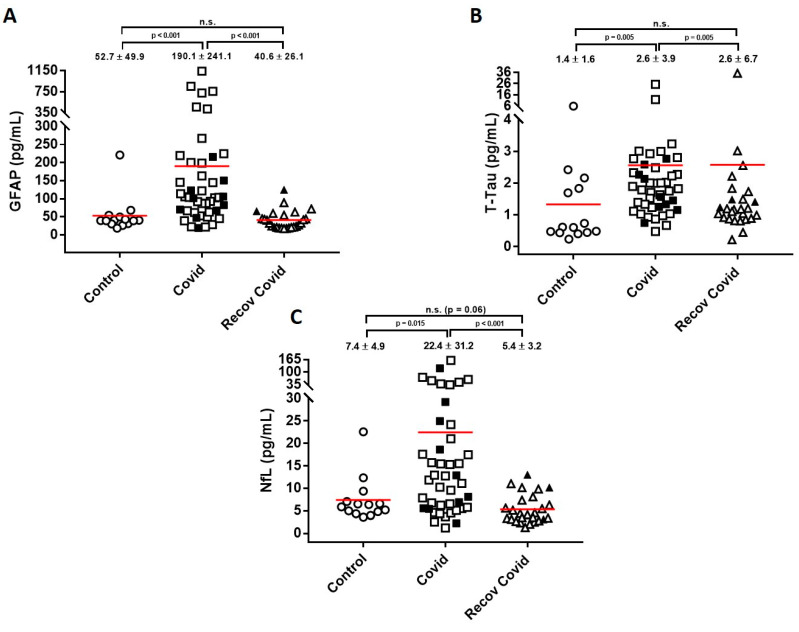
Quantification of astrocytic and neuronal injury biomarkers in COVID-19 patients. The absolute values obtained in plasma for (**A**) glial fibrillary acidic protein (GFAP), (**B**) total tau (T-tau) and (**C**) neurofilament light (NfL) from controls, COVID-19 patients at acute phase of the disease (Covid) and at recovery period (recov Covid; interval of ~2 month between hospital admission and recovery) are shown. Circles: Controls; squares: Covid; triangles: recov Covid; white symbol: moderate severity; black symbol: severe cases. The values are represented as mean, and indicated as mean ± SD. *p* values estimated by Mann–Whitney U test are indicated. Note the segmentation of the Y axis in all 3 graphs.

**Figure 2 ijms-24-02715-f002:**
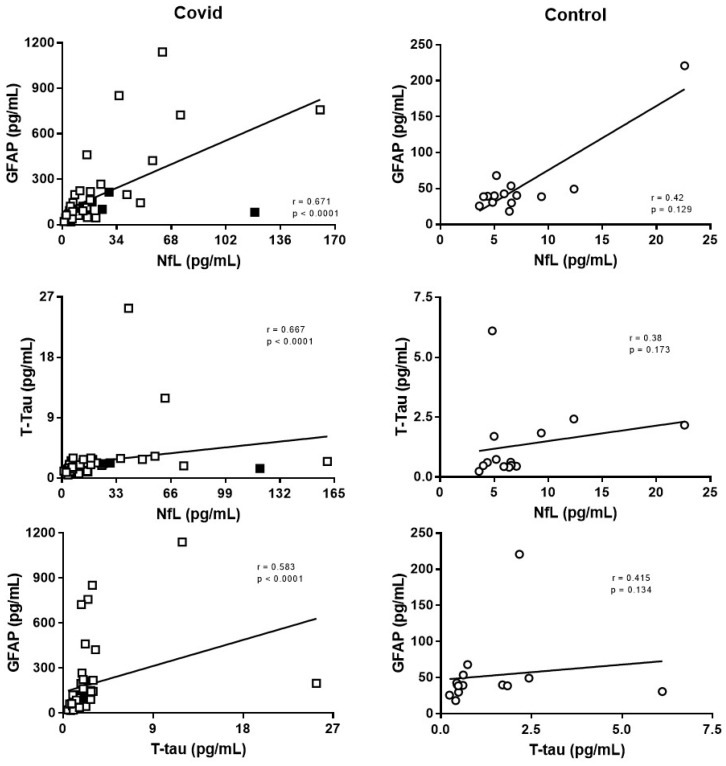
Correlations between plasma GFAP, T-tau and NfL levels across COVID-19 and control groups. A positive correlation was obtained when estimated levels of the biomarkers (see Figure 1) were confronted in COVID-19 in the acute phase of the disease (Covid), but not in controls. Squares: Covid; white symbol: moderate severity; black symbol: severe cases; circles: Controls. Rho (r) and *p* values, estimated by the Spearman test, are displayed.

**Figure 3 ijms-24-02715-f003:**
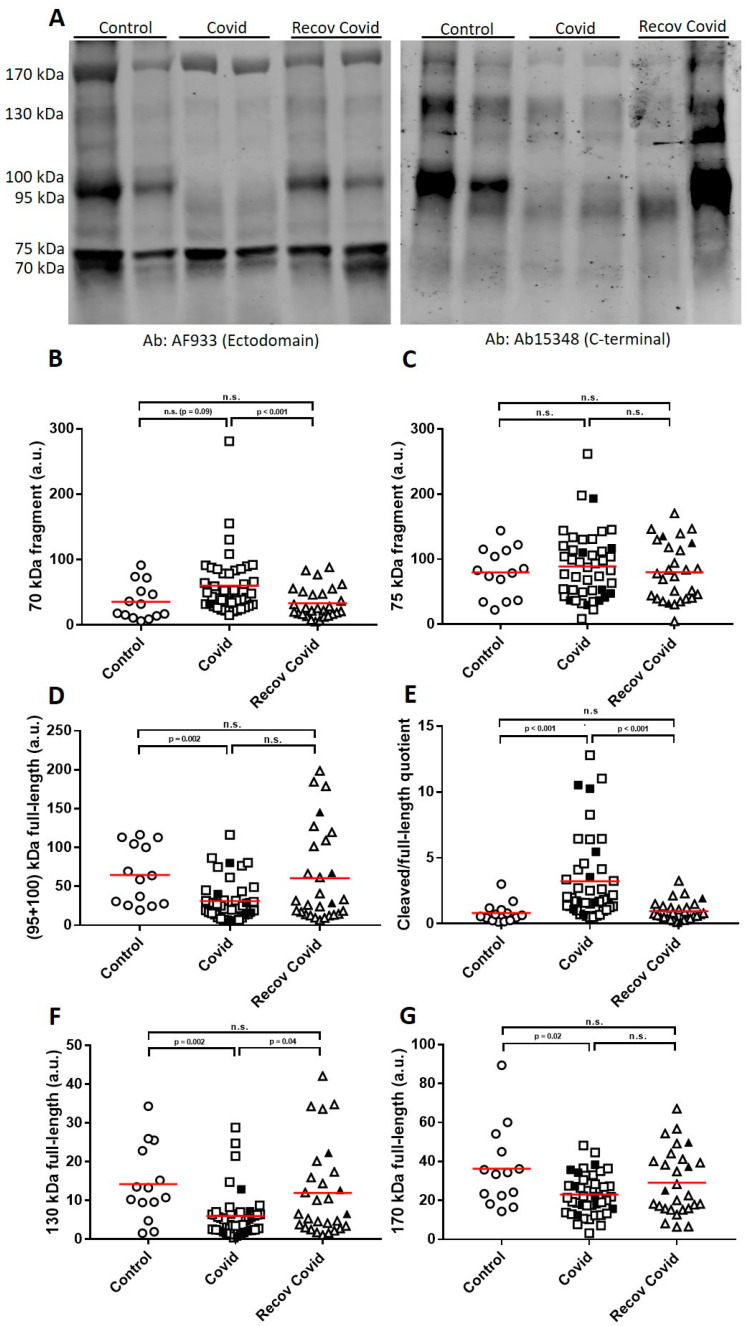
Densitometric quantifications for circulating ACE2 species in COVID-19 patients. (**A**) Representative western blots of control cases, COVID-19 patients at acute phase of the disease (Covid) and at recovery (interval of ~2 month between hospital admission and recovery) were resolved, as indicated, with the AF933 (left image) that recognizes the ectodomain of ACE2, thus recognized all ACE2 species, and ab15348 (right image) raised against the C-terminus of human ACE2, thus detecting only full-length species and not C-terminal truncated fragments. Densitometric quantification of the (**B**) 70 kDa fragment, (**C**) 75 kDa fragment, (**D**) sum of 95 + 100 kDa full-length species, (**E**) the ACE2 cleaved (70 kDa)/full-length (95 + 100 kDa) quotient obtained by dividing the level of immunoreactivity of the 70 kDa fragment by the sum of the levels of immunoreactivity of the 95 and 100 kDa full-length form. Densitometric quantification of the (**F**) 130 kDa full-length species and (**G**) 170 kDa full-length species of ACE2 are shown. Circles: Controls; squares: Covid; triangles: recov Covid; white symbol: moderate severity; black symbol: severe cases. The values are represented as mean. *p* values estimated by Mann–Whitney U test are indicated.

**Table 1 ijms-24-02715-t001:** Characteristics of the COVID-19 patients at the acute phase of infection.

	COVID-19 Patients Characteristics
Demographic and clinical data	
Presentation (moderate/severe)	35/10
Hospitalization (days)	10.5
Diabetes	6.2 (%)
Hypertension	32.3 (%)
Obesity	34.2 (%)
Neurological symptoms	
Headache	26.9 (%)
Memory loss	23.1 (%)
Asthenia	50.0 (%)
Myalgia/arthralgia	26.9 (%)
Anosmia/ageusia	26.9 (%)
Vision impairment	3.8 (%)
Hemaetological parameters	
Lymphocytes (per mm^3^)	1130 [790–1410]
Troponin T (ng/L)	12 [5–21]
CRP (mg/dL)	5.0 [2.4–7.9]
BNP (pg/mL)	109 [37–475]
LDH (U/L)	255 [201–304]
D-dimer (mg/mL)	0.75 [0.40–1.23]
IL-6 (pg/mL)	28 [19–59]
Ferritin (mg/L)	720 [423–1104]

Antecedents of diabetes, hypertension and obesity were considered. Minor neurological symptoms were assessed at re-evaluation after 3 months of the acute episode. None of the patients suffer seizures, stroke, meningitis/encephalitis or dementia. Lymphocytes and plasma markers for heart damage and inflammation such as C-reactive protein (CRP), troponin, B-type natriuretic peptide (BNP), lactate dehydrogenase (LDH), D-dimer, interleukin-6 (IL-6) and ferritin were also determined, and expressed as median [and the 25th and 75th percentiles].

## Data Availability

The data presented in the study are included in the article. Further inquiries can be directed to the corresponding authors.

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
