# Peer review of "Transient Changes in the Plasma of Astrocytic and Neuronal Injury Biomarkers in COVID-19 Patients without Neurological Syndromes"

_ijms, 2023, doi:10.3390/ijms24032715_

Round 1
Reviewer 1 Report
There are a few issues with the statistics and methodology that probably need addressing.
In general, when the statistical tests used are unclear and should be reported both in the text and in the figure legends. Where correlations are mentioned, the graphs should probably be shown in the figure. Also, (not sure if this due to being a peer review copy) but the figure quality is poor.
The COVID and recovered groups are mostly paired (17/26 are the same patients). This may possibly be a problem if and where they are using one-way ANOVAs/Kruskal-Wallis with post-hoc tests for these between group comparisons. In these figures, they show significant differences between the COVID and recovered group means but not report on it in the results text.
Figure 2 is a rework of data from their previous paper (https://faseb.onlinelibrary.wiley.com/doi/10.1096/fj.202100051R , Figure 4) on plasma ACE2 species. Except, the novel aspect here is that they are correlating it with the neuropathology biomarkers (which is mentioned in the text but not shown). Furthermore, it is a bit unclear what the methods are for these densiometric quantifications of the blots. They do not mention how they are normalising the blots - in the previous paper they say with a plasma control sample but they should also include it here. Because, to me, in that representative blot the 70 kDa band looks similar across the groups. Also, I'm not sure if this is because they are plasma samples, but a loading control could have been performed too.
Reviewer 2 Report
The authors described the levels of Nfl, T-Tau, and GFAP in the plasma of COVID-19 patients without major neurological manifestations. I think the manuscript can be improved significantly to have a more sound impact on readers.
1. Define the duration of the acute phase in this study.
2. Are patients with elevated GFAP the same patients with elevated T-tau and Nfl based on the graph presented? and are there correlations based on what is elevated in each patient?
3. Based on the hematological parameters presented in Table 1, were there any correlations found in the literature that show correlations with increased Nfl, T-tau, and GFAP with e.g. increase in IL6 levels? Are there other hematological parameters, cytokines, chemokines, etc that might be more related to GFAP, Nfl, and T-tau in these patients?
4. The ACE2 results do not really enhance the study, since this was already reported and most likely found in COVID-19 whether asymptomatic or symptomatic. And is this not already expected? It's difficult to correlate ACE2 levels found in the plasma with the neurological and axonal markers as ACE2 is not found in the brain.
5. The graphs are a bit blurred (not sure if this is a technical issue during upload).
Reviewer 3 Report
This article clearly and uses an interestingly well-designed to the role of “Transient changes in serum of astrocytic and neuronal injury 2 biomarkers in COVID-19 patients without neurological syn- 3 dromes”. The manuscript has novelty and is noticeable. There are minor points that authors need to consider.
Comments,
1. inclusion and exclusion criteria should for patients selection be added to the material and methods.
2. The authors collected serum or plasma, I am confused, please explain.
“All plasma GFAP, tau and NfL measurements were performed at the Clinical Neurochemistry Laboratory at Sahlgrenska University Hospital by board certified laboratory technicians blinded to clinical data using the Neurology 4-PLEX A assay run on an HD-X Analyzer, as described by the manufacture”
